# Financial Attributes, Environmental Performance, and Environmental Disclosure in China

**DOI:** 10.3390/ijerph17238796

**Published:** 2020-11-26

**Authors:** Die Wu, Shuzhen Zhu, Aftab Ahmed Memon, Hafeezullah Memon

**Affiliations:** 1Glorious Sun School of Business and Management, Donghua University, Shanghai 200051, China; z_shuzhen@dhu.edu.cn; 2College of Public Administration, Nanjing Agricultural University, Nanjing 210095, China; 3College of Textile Science and Engineering (International Institute of Silk), Zhejiang Sci-Tech University, Hangzhou 310018, China

**Keywords:** carbon disclosure information, media reporting, corporate finance, environmental performance

## Abstract

Contest between the international or national enterprises stimulates the formation of innovative or improved products or of well-organized processes. Nevertheless, reliance on carbon-based materials and energy emission sources has been highlighted as a primary problem of the 21st century. The current study examines the influence of carbon disclosure information (CDI), media reporting and financial influence on state-owned enterprises (SOEs) and non-state-owned enterprises (NSOEs) by using Shenzhen and Shanghai’s heavy polluting listed industries’ dataset from 2014 to 2019. By applying different data approaches, the estimated results demonstrate that the CDI level is significantly negative related to SOE compared to NSOE. The estimated results explain that media’s positive reporting offsets the additional benefits to stakeholders. While media’s negative reporting negatively influences a firm’s competitive position, it mitigates the stock price and its social value. Our results suggest that external factors are encouraging for the financial values of stakeholders, along with those of enterprises.

## 1. Introduction

Since energy is the national economy’s basis, climate change is also considered a business issue or a political agenda. For many years, China has been recognized as the world’s leading coal consumer and carbon emitter [1,2]. The measuring of national carbon emissions among trading markets was started in 2017, covering the major industries of China. However, carbon disclosure information is not a yardstick that always reflects the degree of coordination between economic development and energy consumption or emission reduction. This information is considered as the foundation of a carbon emissions trading mechanism, which can advance and greatly improve the individual’s living standard. However, these developments brings problems, such as the extensive economic growth pattern and the inefficient use of energy [3]. Globally, energy issues have attracted attention in several situations of increasingly scarce energy resources and environmental problems, which is important for improving energy practice efficiency to ensure the coordinated development of the economy and energy consumption [4,5]. Domestic scholars [6,7,8,9,10] have conducted studies based on energy efficiency dimensions: efficiency of energy, energy savings, and emission reduction. However, improvement is needed to resolve the conflict between economic development and environmental constraints. China’s rapid economic development and its scarce resources are directly related to energy efficiency [11].

Meanwhile, filling the gap between demand for and supply of energy can effectively promote an economic and sustainable development environment in China. Apparently, domestic and international scholars have paid more attention to investigating higher energy consuming industries [10,12], and comparatively few studies have investigated light industries, such as the textile industries. However, China’s strategy is to save energy, reduce emissions, and improve energy efficiency for the green gross domestic product (GGDP) throughout all industries. This paper selects high-polluting industries in China, for whom primarily information disclosure creates a positive image, rather than focusing on social activities or other responsibilities. Moreover, the research also provides insight based on the financial gains and losses of SOEs and NSOEs caused by CDI reporting.

The primary reason for the disclosure of carbon information in industry is related to external pressure, which comes from the media’s negative reports, state or national pressure, and public awareness, while the media’s positive reporting reflects that enterprises are enacting policies based on rules and regulations from the central or state government. There is no consensus measurement index for carbon information disclosure, which is still exploratory. The government of China has determined that the expected target of CO_2_ will be reduced by around 60% or 65% by 2030 compared to 2005 [13]. At the same time, media is an important source of CDI by investigating corporate governance regarding industries’ high-quality carbon disclosure information, and whether the cost of private equity or venture capital is exaggerated by the quality of CDI, or whether social awareness is fostered by CDI. Media reports increase the influence of CDI on budget, equity financing, and promoting or inhibiting value. For this reason, this study is needed to explain the relationship between CDI, media reports, state or non-state pressure and financial depression on shareholders to make policy recommendations for the improvement of environmental protection and sustainable development.

The paper’s structure is organized as follows: Section 2 presents a literature review, Section 3 presents a theoretical analysis and hypothesis development, Section 4 presents research methods, Section 5 presents results, and the last section presents concluding remarks on this paper’s main findings, conclusion, and limitations.

## 2. Theoretical Literature Review

### 2.1. Environmental Disclosure Theory

The current section explains different methodologies to elucidate the connection between carbon disclosure information and corporate environmental performance [6,14,15,16,17,18,19,20]. Some scholars [21,22] have shifted their curiosity towards emission performance, away from environmental performance’s general characteristics. In addition, a number of economists have also conducted studies to envision the primary determinants of carbon disclosure information (CDI). However, none of them has assessed the CDI level’s effect on the specific SOE and NSOE [23,24]. Recent studies [25,26] have suggested that investors incorporate voluntary carbon information in their pricing decisions. These results explained that GHG emission level is negatively correlated with stock prices, especially from the carbon-intensive industries. On the other hand, measuring GHG emission levels and voluntary disclosure can incur significant costs on the firm; higher exposure can lead to legal action, a threat to the firm’s competitive position, and enhanced scrutiny by regulatory authorities.

Another study from the USA employed the Environmental Input Output Life cycle Assessment Model (IOLCAM) to assess the scope of disclosed carbon emission by the largest industries [27]. The GHG levels were significantly lower in corporate reports compared to carbon disclosure project reports [28]. In addition to the association between internal and external economic factors, researchers have also focused on the influencing factors on carbon disclosure information. Guo et al. [29] used historical data related to carbon emission and GDP from 2005 to 2014 related to China’s multiple cities and found that Chinese cities were different from one another in terms of this long-term association. There were significant modifications in economic structure, developmental mode, and logistical expansion level. Gonzalez et al. [30] exposed the level of carbon disclosure information and its transparency in influencing the market, shareholders, society, and international interaction.

Nevertheless, in the Spanish case, the strongly related factors are shown to be company size, company’s listing order in the IBEX35 and FT500 indexes, financial risk, and ownership concentration. The carbon disclosure factor was a socially critical determinant for the financial market, while institutional and economic factors failed to significantly impact voluntary carbon disclosure [31]. To assess an enterprise’s sustainable growth [7] the leading indices of carbon disclosure information, GHG emission, and carbon intensity were used. The results show that a higher degree of financial development of an enterprise relies on CDI. Moreover, CDI is an internal and external factor, as government pressure and the industry’s sensitivity to the marketing environment have a positive impact on the level of the regional market [32].

However, previous empirical studies have delivered limited support for voluntary disclosure of CDI in the context of media reporting. Dyck and Zingales [33] provided systematic and conclusive evidence that the media influences the company’s policy toward corporate resources and the environment, which is diverted to controlling shareholders’ sole benefits. It should be noted that when enterprises face undesirable incidents, they will use social and environmental reports as a tool to manage their legitimacy [33]. In contrast, the negative impact of the media’s legitimacy can be seen in an environmental press release campaign which did not disclose the annual environmental report [34]. Remarkably, Beatty et al. [35] reported that the capital market responds to corporation disclosure of negative news based on carbon information, but the results did not explain the impact of positive reporting on stakeholders.

A global perspective [36] specifies that broad-spectrum firms with better CSR scores are significantly associated with lower equity capital cost in Europe and North America, but these results are not consistent with Asian countries. Chen et al. [37] argue that assessing disclosed environmental information is beneficial to minimize the investor’s estimation error and equity capital cost. Likewise, it is suggested that carbon management companies positively impact the financial performance of other companies [38]. Bhattacharya et al. [39] reported higher equity financial cost in the stock market related to poor information disclosure from countries with less active trading in stocks. It has been reported that extensive disclosure information could reduce uncertainty for forecasting enterprises [40,41]. Supportively, studies have theoretically proved that investors with low information disclosure bear high risk in stock; meanwhile, there is low demand for small stock and higher financial costs for enterprises [42,43,44]. Therefore, the media’s interference in the provision of CDI supports the researchers’ studies. On the contrary, media’s positive or negative reporting has financial impact on a firms’ social value.

Multiple studies have examined the impact of carbon disclosure information and its economic consequences. It has been reported that enterprises’ environmental agenda and media agenda are a mirror of each other, while some impacts are different, but not vice versa [45]. Still, a study is needed to understand CDI’s systematic influence on state-owned enterprises (SOE) and non-state-owned enterprises (NSOE), primarily because of the CDI quality effect on various stakeholders’ market competitiveness position. Moreover, previous studies have focused on the macro-environment, without classifying the carbon disclosure information nor the impact of the media’s positive or negative reporting on CDI and the media’s interaction with SOEs and NSOEs. Carbon information disclosure can be divided into financial and non-financial influencing factors. The above studies focused only on uni-directional causalities; less research was conducted on media reporting and the financial impact on SOEs or NSOEs of carbon disclosure information.

### 2.2. Empirical Literature Review and Hypothesis Development

The public pressure on CDI connected to corporations primarily comes from media reporting, shareholders, and governmental authorities. Therefore, enterprises are required to voluntarily disclose carbon information before the annual survey, which is important for the government to estimate and make policies accordingly for improving environmental protection and protecting the ecological environment.

Since 1979, the Chinese government has enacted environmental protection laws regarding air pollution, land pollution, water pollution, and other relevant regulations. However, from June 2012, the National Development Reform Commission voluntarily issued a “Management for Transactional Measurement” to reduce greenhouse gas emission transactions. In addition, the Ministry of Finance’s regulations also deal with disclosure of companies’ environmental information and its inclusion in primary accounting standards of the firm: contingent events, auditing standards for certified public accounts no. 1631, financial auditing statements or issues related to the environment. Meanwhile, the China Bank Regulatory Commission (CBRC) has issued the Green Credit Guidelines (GCRs) to take full advantage of banks in promoting energy conservation, reducing carbon emission, and promoting environmental protection. Besides, Green Credit (GC), Green Insurance (GI), Green Securities (GS), and other laws and regulations were enacted one after another.

Of course, a series of laws and regulations enforced by the government and other regulatory departments creates pressure for companies and their on CDI. For example, the listed environmental protection department classifies or inspects industries that are emitting heavy pollutants. The China Securities Regulatory Commission (CRSC) has imposed stringent regulations on disclosing environmental information to companies listed in heavily polluting industries. To create public awareness, the media is also responsible for exposing companies’ CDIs through positive or negative reporting. On the other hand, the media’s negative reporting on companies’ CDI puts pressure on the heavily polluting industries, mitigates capital value, and extenuates stock price. We cannot deny the systematic competition among media companies to enhance channel ratings, whether positive or negative. However, China’s whole media is controlled by the central government. Therefore, the employed data source and updated data set are reliable to estimate the hypotheses below.

**Hypothesis 1** **(H1).**
*There is a strong positive relationship between Government pressure and carbon information disclosure.*


In China, the government has promulgated environmental policies to improve long life expectancy through a healthy provision for the environment. For example, central government enforcement is considered a major influential factor on stakeholders because their supervisory power has forced corporate economic-driven strategies to develop those related to environmental and societal welfare. However, the state departments also have some social responsibilities and sometimes remain under pressure from the central government department regarding carbon disclosure. Therefore, the SOE’s administrative characteristics remain limited due to the policy burden. Brambilla et al. [46] suggested that this bureaucratic structure can improve the firm’s disclosure level of environmental information because enterprises face several pressures: media reporting, political burden, or state laws and regulation systems. In other words, stakeholders’ perception is that government pressure is directly proportionate to carbon disclosure information. This pressure tends to economic loss or threat to the firm’s competitive position. In addition to media interest, the below assumption is made.

**Hypothesis 2** **(H2).**
*The frequency of positive media reports based on carbon disclosure information has a positive impact on enterprises.*


**Hypothesis 3** **(H3).**
*The frequency of negative media reports based on carbon disclosure information has a negative impact on enterprises.*


### 2.3. Carbon Information Disclosure and External Pressure

Socio-political theory shows that media or corporations voluntarily disclose carbon information, which may tend to bring political or social pressure to a firm [18,47,48,49]. However, enhancing the number of international and regional programs poses challenges to the company’s growing compliance risks. Media pressure has become one of the primary factors shaping enterprises’ strategies regarding carbon disclosure and environmental protection (see Figure 1). The empirical evidence [50,51] has documented that private firms are more sensitive than SOEs regarding the association between media and regulatory pressure, so new actions need to be implemented.

**Hypothesis 4** **(H4).**
*The interaction between the media’s positive reporting in state-owned enterprises or non-state-owned enterprises will benefit the corporation.*


**Hypothesis 5** **(H5).**
*The interaction between media’s negative reporting and the state-owned enterprise or non-state-owned enterprise will result in mutual damage to the corporation.*


To address the above hypothesis, interaction between media and corporation is based on carbon, which is measured as the extent to which a firm voluntarily discloses carbon information to the public through the media before its annual official disclosure of carbon information, which means that frequent carbon disclosure may offset the negative influences of CDI on firm value. In other words, the media-independent communication approach can extenuate negative stock price shocks affected by disclosure of CDI through media negative reporting. Moreover, media coverage and visibility may increase the publication of enterprises’ supplementary claims [52]. On the other hand, independent carbon disclosure through the media may offset the positive effects of CDI on firm value.

## 3. Research Design

The current paper uses a data sample of Chinese listed companies on the Shanghai Stock Exchange (SSE) and the Shenzhen Stock Exchange (SZSE) from 2014 to 2019. The selection procedure follows the following restrictions. First, insurance companies and financial departments were restricted, Second, companies’ data were eliminated containing missing information in other variables. Certain features were restricted to estimate the intended hypothesis; the final data sample included 14,159 companies’ observations. Carbon disclosure information and media (positive or negative) reported information is obtained from the Chinese Research Data Services (CNRDs). We treated the tail with a continuous variable at 1% and a 99% level by eliminating outsiders’ influence.

### 3.1. Variable Explanations and Methodology

Media reporting: This is classified into sub-categories: positive, negative, and neutral reporting. Positive reporting involves an organization’s environmental protection activities; negative reporting is related to creating environmental pollution; neutral reporting includes implementing policies nationally and industries following rules and regulations regarding environmental protection.

Carbon information disclosure: According to past studies [17,34] this is divided carbon into financial and non-financial carbon disclosure information. Therefore, the current paper uses the actual description of Chinese listed companies’ social responsibilities and environmental disclosure reports. It is divided into five different aspects: accounting of carbon emission reduction, financial inputs, carbon emission performance, environmental accidents, and government subsidies. Using sub-categories, Table 1 describes secondary indicators with definitions.

Control variables: We have used additional variables to control the model’s misspecification, directly or indirectly affecting and MR (Media Reporting negative/positive (MNR/MPR)). The control variables are composed of leverage finance (LF), natural LN size (LN Size), market low (ML), market high (MH), turnover rate (TOR), rate of assets (ROA), fixed assets ratio (FAR), age, market-book ratio (MB), income growth rate (IGR) and foreign shareholders (FSH).

### 3.2. Models

To test the relationship between carbon information disclosures, media reporting, and government pressure, we have used the following models
(1)$$\begin{array}{ll} CDI_{,\;i,t}=\alpha + & MPR_{,\;i,t}+\;MNR_{,\;i,t}+SOE_{,\;i,t}+\;NSOE_{,\;i,t}+LF_{,\;i,t}+\;LNSize_{,\;i,t}+\;ML_{,\;i,t}\\ & +\;MH_{,\;i,t}+TOR_{,\;i,t}+\;ROA_{,\;i,t}\;+\;FAR_{,\;i,t}+\;Age_{,\;i,t}+\;MBR_{,\;i,t}\\ & +IGR_{,\;i,t}+FSH_{,\;i,t}+\;\varepsilon \end{array}$$
where the dependent variable is CDI presents the level of carbon information disclosure. *i* denotes index (*i* = 1 … N), and *t* is the period of index (*t* = 2014 to 2019). The state-owned enterprises (SOEs) and non-state-owned enterprises (NSOEs) are related to heavily polluting industries considered dummy variables. Media negative reporting (MNR) and media positive reporting (MPR) show the impact of reporting on SOE and NSOE based on carbon disclosure information.
(2)$$\begin{array}{ll} CDI_{,\;i,t}=\alpha \;+ & MPR_{,\;i,t}+\;MNR_{,\;i,t}+SOE_{,\;i,t}+\;NSOE_{,\;i,t}+{\left(NSOE\times \;MPR\right)}_{,\;i,t}\\ & +\;{\left(SOE\times MPR\right)}_{,\;i,t}+LF_{,\;i,t}+\;LNSize_{,\;i,t}+\;ML_{,\;i,t}+\;MH_{,\;i,t}+TOR_{,\;i,t}\\ & +\;ROA_{,\;i,t}\;+\;FAR_{,\;i,t}+\;Age_{,\;i,t}+\;MBR_{,\;i,t}+IGR_{,\;i,t}+FSH_{,\;i,t}+\;\varepsilon \end{array}$$
(3)$$\begin{array}{ll} CDI_{,\;i,t}=\alpha \;+ & MPR_{,\;i,t}+\;MNR_{,\;i,t}+SOE_{,\;i,t}+\;NSOE_{,\;i,t}+{\left(NSOE\times \;MNR\right)}_{,\;i,t}\\ & +\;{\left(SOE\times MNR\right)}_{,\;i,t}+LF_{,\;i,t}+\;LNSize_{,\;i,t}+\;ML_{,\;i,t}+\;MH_{,\;i,t}\\ & +TOR_{,\;i,t}+\;ROA_{,\;i,t}\;+\;FAR_{,\;i,t}+\;Age_{,\;i,t}+\;MBR_{,\;i,t}+IGR_{,\;i,t}\\ & +FSH_{,\;i,t}+\;\varepsilon \end{array}$$

In addition to seeking the interaction between SOEs and NSOEs with MPR and MNR, the results are estimated as a separate column-wise equation.

Arguably, an endogeneity problem can exist in the above described model setting [53]. Following the above model setting as a preliminary experiment, we employed the Hausman specification test to detect the endogenous regressor in a regression model.

## 4. Empirical Results and Discussion

### 4.1. Descriptive Statistics

Table 2 presents the descriptive statistics of the included variables. Table 2 presents the mean and standard value of the dependent variables, independent variables, and control variables. The dependent variable (CDI) presents listed companies’ voluntary disclosure level in China, relevant to the previous study [11]. The mean value of independent variables (SOE/NSOE) and (MNR/MPR) were 0.61/0.37 and 0.65/0.40, respectively, shown in the listed companies. All variables are described with standard deviation, variance, and skewness with reasonable limits.

### 4.2. Pearson Coefficient Correlation

Table 3 shows the Pearson coefficient correlation between the dependent variable, independent variables, and control variables. The Pearson coefficient results show the strength of all variables. The correlation of CDI (dependent variable) is significantly positive with government, non-government, MPR, and MNR at the two-tailed level (0.01). Meanwhile, external, and internal pressure positively influence the disclosure of carbon information. Therefore, we can say that all the variables are significantly correlated with CDI by confirming all the intended hypotheses.

### 4.3. Regression Analysis

Table 4 presents the multiple regression analysis of ANOVA, consisting of a calculation that provides the variability within a regression model and significance level. The residuals $Y-\hat{Y}$ show significant variation between predicted and actual values explained by SOE and NSOE variables. The mean square error term is almost equivalent within SOE and NSOE, indicating equivalent deviation between the observed and fitted values. The *p*-value for the F-test statistic is less than 0.001. Meanwhile, results provide strong evidence against the null hypothesis. The squared multiple correlations for SOE R2 = 192.478/1558.554 = 0.1234, show 12.3% variability in the CDI variable explained by the included prescribed variables. However, for NSOE R2 = 174.609/1010.881 = 0.1727, indicating that the prescribed variables explain 17.2% variability in the CDI variable. Moreover, the degree of freedom (DF) shows 13 independent variables, including control variables, for the regression model.

Table 5 shows the estimated regression results of explanatory variables with the general impact of response variables over heavy polluting SOEs and NSOEs, significantly correlated at the 0.001 level. The regression coefficient of MPR (0.018, 0.011) and MNR (−0.019, −0.015) are significantly correlated in both SOE and NSOE columns, meaning that MPR or MNR affects carbon disclosure information, either putting pressure on the stated financial value or threatening the firm’s competitive position. Meanwhile, SOEs or NSOEs should disclose their carbon information; otherwise, enterprises lose their financial and social strength due to media negative reporting. Hence all the below results support H1, H2, and H3.

### 4.4. Interaction between SOE Pressure and NSOE Pressure on Carbon Information Disclosure

The interaction between enterprises and media reporting based on the disclosure of carbon information is shown in Table 6. The first column shows the regression results of all variables in an ordinary format. The second column results show an insignificant effect of interaction between MPR × SOE and MPR × NSOE based on the disclosure of carbon information at a 10% level, meaning that MPR can lessen or intensify the negative association between SOE and NSOE; however, the consequence is unobservable. Table 6, the third column, shows the significant negative effect of the interaction between MNR × SOE and MNR × NSOE based on the disclosure of carbon information at a 10% level. It seems that the media’s negative reporting affects more SOEs compared to NSOEs; our results contrast with a previous report that heavy pollution is caused by SOEs [11]. Hence all the below results support H1, H3, and H5, but not H2 and H4.

### 4.5. Control of Heteroscedasticity

In order to justify the heterogeneity or individuality among companies by allowing them their intercept, as the intercept may differ across companies, it does not vary over time because it is invariant. Table 7 explains the employed panel data time series across 101 companies to explain the Hausman test’s variant difference. We did find significant results among the Fixed Effects model ($\hat{\beta }FE$) and Random Effects model ($\hat{\beta }RE$). The results of the Hausman test difference (b-B) probability were below *p* < 0.05, which explains that the fixed effect model has been accepted and the random effect model is soundly rejected.

## 5. Conclusions

The study reviews the relationship of environmental exposure and its interaction with media reporting, and its financial impact on listed companies of SSE, SZSE and CNRDS from 2014 to 2019. According to CNRDS data, the results show that state-owned enterprises voluntarily disclose a higher carbon information level, while non-state-owned enterprises are revealing a lower level of carbon information. According to stakeholder’s theory, enterprises should understand their social and political responsibilities before making business decisions based on their self-interest or accepting constraints.

State-owned enterprises are heavily polluting enterprises based on the level of carbon information disclosure, higher than non-state-owned enterprises. In addition, we have also found that NSOEs are enjoying more premiums compared to SOEs for two reasons: either NSOEs are voluntarily disclosing CDI or they are enacting the rules and regulations of central government, and that is why NSOEs are leading in the market competition. Moreover, the interaction of media with SOEs and NSOEs also supports the above statement. Briefly, a firm’s reaction to voluntary disclosure of carbon information is not favorable to investors. In other words, the media-independent communication approach can extenuate negative stock price shocks affected by disclosure of CDI through media negative reporting. On the other hand, independent carbon disclosure through media may offset the positive effects of CDI on a firm’s value. Therefore, firms should understand that carbon disclosure can quickly become de facto regulation, and it will be difficult to detect or avoid in the near future.

Although the current study has taken into account a large amount of data and employed a common technique for sorting out missing information to estimate the intended hypothesis, some limitations exist. Data was collected from different sources. It would support the researchers or think-tankers if enterprises voluntarily disclosed appropriate information to the public through media sources in order to obtain public support and prove that their behavior conforms to social values, thereby maintaining legitimacy. However, companies disclose information by acting upon the state rules and regulations. Sometimes managers face myriad challenges. Therefore, they prioritize their social and personal benefits instead of providing voluntary services to protect the environment. Second, researchers recommend employing other indicators to assess enterprises’ financial performance to bring more conclusive findings to the literature.

## Figures and Tables

**Figure 1 ijerph-17-08796-f001:**
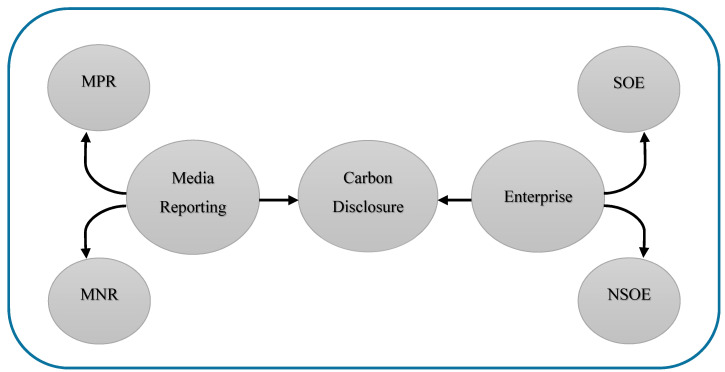
Interaction between enterprises’ carbon disclosure information and media reporting. Acronyms: Media Positive Reporting (MPR); Media Negative Reporting (MNR); State-Owned Enterprise (SOE); Non-State-owned Enterprise (NSOE).

**Table 1 ijerph-17-08796-t001:** Definitions of Variables.

Variables	Names	Symbol	Definitions
CDI	Carbon Disclosure Information	+	CDI Carbon Disclosure Information = 1; Carbon Information Not Disclosed = 0
MNR	Media Negative Reporting	+	Denotes the number of times negatively reported by media
MPR	Media Positive Reporting	+	Denotes the number of times positively reported by media
SOE	State-Owned Enterprise	+	Governmental Pressure used as a Dummy variable; State-owned enterprise = 1; non-state-owned = 0
NSOE	Non-State-owned Enterprise	+	Governmental Pressure used as a Dummy variable; non-state-owned = 1; state-owned = 0
LF	Leverage Finance	+	Asset-Liability Ratio = Total Liabilities/Total Assets
LN Size	Natural LN Size	+	Denotes the logarithm of the total book value of the assets of the company at the end of the year
ML	Market Low	+/−	Provincial lowest average of marketization index per year
MH	Market High	+/−	Provincial highest average of marketization index per year
TOR	Turnover Rate	+	Sum of the turnover rate of tradable shares the current year
ROA	Rate of Assets	+/−	Return on Assets = Net profit/Average Total Assets
FAR	Fixed Assets Ratio	+/−	Fixed Asset Ratio = Fixed assets/Total Assets
Age	Age	+/−	Number of years passed since the company’s Initial Public Offering (IPO—Initial Public Offering) time
MBR	Market-book ratio	+/−	the market value/the book value
IGR	Income Growth Rate	+/−	Income growth rate
FSH	Foreign Shareholders	+/−	Percentage of foreign shares within the company

**Table 2 ijerph-17-08796-t002:** Descriptive Statistics.

Variables	Minimum	Maximum	Sum	Mean	Std. Deviation	Variance	Skewness	
	Statistic	Statistic	Statistic	Statistic	Statistic	Statistic	Statistic	Std. Error
Carbon disclosure information (CDI)	0	1	10,503	0.74	0.438	0.192	−1.105	0.021
Media Negative Reporting (MNR)	0	21	9242	0.65	1.169	1.367	7.101	0.021
Media Positive Reporting (MPR)	0	19	5658	0.40	0.839	0.704	5.389	0.021
State-Owned Enterprise (SOE)	0	1	8663	0.61	0.487	0.238	−0.459	0.021
Non-State-Owned Enterprise (NSOE)	0	1	5179	0.37	0.482	0.232	0.557	0.021
Leverage finance (LF)	0	1	9161	0.65	0.478	0.228	−0.615	0.021
Natural Log Size (LN Size)	8	30	321,585	22.71	3.687	13.590	−0.091	0.021
Market low (ML)	0	1	5912	0.42	0.493	0.243	0.334	0.021
Market high (MH)	0	1	8211	0.58	0.494	0.244	−0.324	0.021
Turnover rate (TOR)	0.0100	24.46	91,346.35	6.451470	5.4317	29.504	0.589	0.021
Rate of assets (ROA)	−0.4927	0.983	292.13	0.020633	0.270	0.073	0.120	0.021
Fixed assets ratio (FAR)	0.0023	1.0255	3919.86	0.276846	0.1852546	0.034	0.202	0.021
Age	0.0000	27.480	115,532.47	8.159649	4.32274	18.686	0.363	0.021
Market-Book Ratio (MBR)	0.0000	18.049	89,536.95	6.323678	5.52741	30.552	0.652	0.021
Income growth rate (IGR)	−0.97	1.92	5884.74	0.4156	1.256	1.580	0.070	0.021
Foreign shareholders (FSH)	0.0594	0.996	7608.19	0.537340	0.26382	0.070	−0.051	0.021
Valid N (list-wise)	14,159							

**Table 3 ijerph-17-08796-t003:** Pearson Correlation Coefficient.

Variables	CDI	MNR	MPP	SOE	N/SOE	LF	LN Size	ML	MH	TOR	ROA	FAR	Age	MBR	IGR	FSH
CDI	1	0.960 **	0.162 **	0.076 **	−0.012	−0.007	−0.080 **	−0.096 **	0.100 **	−0.006	0.208 **	−0.179 **	−0.013	−0.091 **	−0.001	0.056 **
MNR	0.960 **	1	0.198 **	−0.065 **	0.086 **	−0.064 **	−0.162 **	−0.108 **	0.109 **	−0.036 **	−0.097 **	−0.206 **	−0.138 **	−0.082 **	−0.028 **	0.015
MPR	0.162 **	0.198 **	1	−0.060 **	0.066 **	−0.055 **	−0.125 **	−0.108 **	0.109 **	−0.025 **	−0.041 **	−0.139 **	−0.140 **	−0.051 **	−0.014	0.036 **
SOE	0.076 **	−0.065 **	−0.060 **	1	−0.953 **	0.884 **	0.216 **	−0.158 **	0.161 **	0.128 **	−0.042 **	0.064 **	0.340 **	−0.377 **	0.050 **	−0.104 **
NSOE	−0.012	0.086 **	0.066 **	−0.953 **	1	−0.927 **	−0.265 **	0.161 **	−0.164 **	−0.154 **	0.014	−0.118 **	−0.367 **	0.414 **	−0.077 **	0.129 **
LF	−0.007	−0.064 **	−0.055 **	0.884 **	−0.927 **	1	0.263 **	−0.175 **	0.180 **	0.161 **	−0.021 *	0.111 **	0.368 **	−0.437 **	0.090 **	−0.125 **
LN Size	−0.080 **	−0.162 **	−0.125 **	0.216 **	−0.265 **	0.263 **	1	−0.140 **	0.139 **	−0.183 **	0.172 **	0.458 **	0.402 **	−0.079 **	−0.063 **	−0.036 **
ML	−0.096 **	−0.108 **	−0.108 **	−0.158 **	0.161 **	−0.175 **	−0.140 **	1	−0.992 **	0.034 **	0.199 **	0.322 **	0.130 **	0.674 **	0.059 **	−0.029 **
MH	0.100 **	0.109 **	0.109 **	0.161 **	−0.164 **	0.180 **	0.139 **	−0.992 **	1	−0.033 **	−0.204 **	−0.321 **	−0.132 **	−0.678 **	−0.061 **	0.026 **
TOR	−0.006	−0.036 **	−0.025 **	0.128 **	−0.154 **	0.161 **	−0.183 **	0.034 **	−0.033 **	1	0.063 **	0.038 **	0.009	−0.099 **	0.433 **	−0.175 **
ROA	0.208 **	−0.097 **	−0.041 **	−0.042 **	0.014	−0.021 *	0.172 **	0.199 **	−0.204 **	0.063 **	1	0.220 **	0.139 **	0.162 **	0.047 **	−0.014
FAR	−0.179 **	−0.206 **	−0.139 **	0.064 **	−0.118 **	0.111 **	0.458 **	0.322 **	−0.321 **	0.038 **	0.220 **	1	0.379 **	0.180 **	−0.044 **	−0.063 **
Age	−0.013	−0.138 **	−0.140 **	0.340 **	−0.367 **	0.368 **	0.402 **	0.130 **	−0.132 **	0.009	0.139 **	0.379 **	1	−0.160 **	−0.050 **	−0.146 **
MBR	−0.091 **	−0.082 **	−0.051 **	−0.377 **	0.414 **	−0.437 **	−0.079 **	0.674 **	−0.678 **	−0.099 **	0.162 **	0.180 **	−0.160 **	1	−0.025 **	0.096 **
IGR	−0.001	−0.028 **	−0.014	0.050 **	−0.077 **	0.090 **	−0.063 **	0.059 **	−0.061 **	0.433 **	0.047 **	−0.044 **	−0.050 **	−0.025 **	1	−0.186 **
FSH	0.056 **	0.015	0.036 **	−0.104 **	0.129 **	−0.125 **	−0.036 **	−0.029 **	0.026 **	−0.175 **	−0.014	−0.063 **	−0.146 **	0.096 **	−0.186 **	1

**—Correlation is significant at the 0.01 level (two-tailed). *—Correlation is significant at the 0.05 level (two-tailed). CDI—Carbon Disclosure Information; MNR—Media Negative Reporting; MPR–Media Positive Reporting; SOE—State-Owned Enterprise; NSOE—Non-State-owned Enterprise; LF—Leverage Finance; LN Size—Natural LN Size; ML—Market Low; MH—Market High; TOR—Turnover Rate; ROA–Rate of Assets; FAR—Fixed Assets Ratio; MBR—Market-book ratio; IGR—Income Growth Rate; FSH—Foreign Shareholders.

**Table 4 ijerph-17-08796-t004:** ANOVA Results.

Variables	Model	Sum of Squares	Df	Mean Square	F	Sig.
SOE	Regression	192.478	13	14.806	93.741	0.000
	Residual	1366.075	8649	0.158		
	Total	1558.554	8662			
NSOE	Regression	174.609	13	13.431	82.956	0.000
	Residual	836.272	5165	0.162		
	Total	1010.881	5178			

Dependent Variable: CDI; Selecting only cases for which SOE = 1; NSOE = 1.

**Table 5 ijerph-17-08796-t005:** Estimated regression results.

Variables	SOE	NSOE
	Coefficients (Std. Error)	Coefficients (Std. Error)
Intercept	−0.808 (0.096) ***	−0.741 (0.102) ***
MPR	0.018 (0.006) ***	0.011 (0.005) **
MNR	−0.019 (0.005) ***	−0.015 (0.005) ***
LF	−0.065 (0.034)	−0.189 (0.028) ***
LN Size	−0.010 (0.001) ***	−0.015 (0.004) ***
ML	0.355 (0.084) ***	0.271 (0.071) ***
MH	0.394 (0.083) ***	0.410 (0.073) ***
TOR	−0.008 (0.001) ***	0.006 (0.001) ***
ROA	0.441 (0.016) ***	0.671 (0.023) ***
FAR	−0.360 (0.025) ***	−0.295 (0.090) ***
Age	0.004 (0.001) **	0.006 (0.002) ***
MBR	−0.009 (0.001) ***	−0.006 (0.003) *
IGR	0.009 (0.004) ***	0.010 (0.005) **
FSH	0.056 (0.018) ***	0.063 (0.021) ***
Number of Obs.	14,159	14,159
R^2^	0.123	0.173

Selecting only cases for which SOE = 1; NSOE = 1; Media positive reporting (MPR), Media negative reporting (MNR), leverage finance (LF), natural LN size (LN Size), market low (ML), market high (MH), turnover rate (TOR), rate of assets (ROA), fixed assets ratio (FAR), market-book ratio (MBR), income growth rate (IGR) and foreign shareholders (FSH). Significance level *** at 1%, ** at 5% and * at 10% level.

**Table 6 ijerph-17-08796-t006:** Interaction between Government and Non-Government with the relationship of MPR and MNR.

Variables	Coefficients (Std. Error)	Coefficients (Std. Error)	Coefficients (Std. Error)
Intercept	0.173 (0.068) ***	0.178 (0.068) ***	0.175 (0.068) ***
MPR	0.018 (0.004) ***	0.038 (0.007) ***	0.016 (0.004) ***
MNR	−0.010 (0.003) ***	−0.009 (0.003) ***	0.022 (0.008)
SOE	0.627 (0.024) ***	0.628 (0.024) ***	0.650 (0.024) ***
NSOE	0.472 (0.030) ***	0.469 (0.030) ***	0.490 (0.030) ***
NSOE * MPR		−0.015 (0.006)	
SOE * MPR		−0.021 (0.005)	
NSOE * MNR			−0.036 (0.010) ***
SOE * MNR			−0.045 (0.010) ***
LF	−0.151 (0.019) ***	−0.153 (0.019) ***	−0.151 (0.019) ***
LN Size	−0.007 (0.001) ***	−0.007 (0.001) ***	−0.008 (0.001) ***
ML	0.314 (0.054) ***	0.314 (0.054) ***	0.314 (0.054) ***
MH	0.365 (0.054) ***	0.365 (0.054) ***	0.367 (0.054) ***
TOR	−0.002 (0.001) *	−0.002 (0.001) *	−0.002 (0.001) *
ROA	0.483 (0.013) ***	0.484 (0.013) ***	0.484 (0.013) ***
FAR	−0.366 (0.024) ***	−0.366 (0.024) ***	−0.367 (0.024) ***
Age	0.005 (0.001) ***	0.005 (0.001) ***	0.004 (0.001) ***
MBR	−0.008 (0.001) ***	−0.008 (0.001) ***	−0.008 (0.001) ***
IGR	0.006 (0.003)	0.006 (0.003)	0.006 (0.003)
FSH	0.081 (0.013) ***	0.082 (0.013) ***	0.079 (0.013) ***
Number of Obs.	14,159	14,159	14,159
Adjusted R^2^	0.162	0.163	0.163

Media positive reporting (MPR), Media negative reporting (MNR), State-Owned Enterprise (SOE); Non-State-owned Enterprise (NSOE); leverage finance (LF), natural LN size (LN Size), market low (ML), market high (MH), turnover rate (TOR), rate of assets (ROA), fixed assets ratio (FAR), market-book ratio (MBR), income growth rate (IGR) and foreign shareholders (FSH). The significance level *** at 1% and * at 10% level.

**Table 7 ijerph-17-08796-t007:** Financial impact of CDI on companies SOE/NSOE and MPR/MNR.

Variables	Fixed Effects (b)	Random Effects (B)	Hausman Test
	Coef. (Std. Error)	Coef. (Std. Error)	$\mathrm{Difference}\;(b-B)\;\surd \left(diag\left(V_{b}-V_{B}\right)\right)$
Intercept	−0.2771 (0.308) ***	−0.3028 (0.282) **	
MPR	−0.0817 (0.013) ***	−0.0758 (0.012) ***	−0.0058 (0.005)
MNR	0.137 (0.022) ***	0.1437 (0.020) ***	−0.0058 (0.007)
SOE	0.4014 (0.091) ***	0.4420 (0.079) ***	−0.0405 (0.045)
NSOE	0.1725 (0.021) ***	0.1730 (0.019) ***	−0.0005 (0.008)
LF	−0.4262 (0.092) ***	−0.4425 (0.080) ***	0.0162 (0.044)
LN Size	0.0223 (0.005) ***	0.0195 (0.004) ***	0.0027 (0.002)
ML	0.7447 (0.275) ***	0.8163 (0.250) ***	−0.0716 (0.114)
MH	0.7599 (0.276) ***	0.8267 (0.252) ***	−0.0668 (0.113)
TOR	−0.0072 (0.003) *	−0.0109 (0.003) ***	0.0037 (0.001)
ROA	0.4446 (0.069) ***	0.3933 (0.062) ***	0.0512 (0.030)
FAR	−0.5837 (0.114) ***	−0.5295 (0.101) ***	−0.0542 (0.052)
Age	0.0035 (0.004)	0.0011 (0.003)	0.0023 (0.002)
MBR	0.0008 (0.004) *	0.0006 (0.003) *	0.0001 (0.001)
IGR	0.0261 (0.012) **	0.04110 (0.011) ***	−0.0149 (0.004)
FSH	−0.0741 (0.054) ***	−0.0342 (0.047) ***	−0.0398 (0.025)
F (15,490)	17.45 ***		
Wald chi2 (15)		293.71 ***	
R-sq.			
Within	0.3482	0.3411	
Between	0.2223	0.2906	
Overall	0.3252	0.3324	
Number of obs.	606		

Media positive reporting (MPR), Media negative reporting (MNR), State-Owned Enterprise (SOE); Non-State-owned Enterprise (NSOE); leverage finance (LF), natural LN size (LN Size), market low (ML), market high (MH), turnover rate (TOR), rate of assets (ROA), fixed assets ratio (FAR), market-book ratio (MBR), income growth rate (IGR) and foreign shareholders (FSH). The significance level *** at 1%, ** at 5% and * at 10% level.

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
