# Peer review of "Financial Attributes, Environmental Performance, and Environmental Disclosure in China"

_ijerph, 2020, doi:10.3390/ijerph17238796_

Round 1

Reviewer 1 Report

The paper under title "Financial Attributes, Environmental Performance, and Environmental Disclosure of China" deals with the determinants of carbon disclosure information on the Chinese market. The paper could be of local interest to researchers on the Chinese market, however it has several flaws making it unsuitable from publication in this current form. I urge the authors to consider the following comments and improve the paper for potential future endeavours.

1) On page 2, lines 58-61 authors have provided a very confused discussion on some questions regarding venture capital or private equity etc. What do they mean exactly? Are these questions examined on this study? As I kept reading it seemed that author do not examine such issues mentioned on the introduction so why are they put forth at the first place?

2)There is not sufficient justification and theoretical background for the scope of the study. This fact makes the contribution of the study very limited to existing literature, even within chinese studies.

3) On page 3, lines 124-125, authors say that they focus on SOEs and non-SOEs as a potential determinant of carbon disclosures. Yet there is not explanation why authors expect such difference at the first place. There is no justification as to why state ownership or not affects CDI.

4) All research hypotheses are not substantiated and even H2 and H3 are not derived from studies or theories related to the research question. Also as hypothesis are stated is very difficult to test and infer if they are valid or not. For example when authors say that CDI has a negative impacrt on enterprises, what "impact" they refer to? They need to be specific. Also the discussion leading to H1 is not supported by other studies. Simillarly, H4 and H5 refer to "benefitis" and "damage" to the corporation. What benefit and what damage? Overall the hypothesis development section needs a lot of work in order to be sound and substantiated.

5) On section 3.1 how positive reporting is separated from neutral? If a firm is following existing environmental rules and regulations isn't it positive? Why s deemed neutral? This distinction requires further consideration. 

6) On page 6, on the paragraph of CID data collection, there are several metrics that are not presented on the first table as suggested. Also this part requires more justification from related literature. 

7) Since CDI is a dummy variable authors should have employed logistic regression analysis and and not panel as performed. This is a serious issue that may affect results and conclusions.

Author Response

Comment # 1 The paper under title "Financial Attributes, Environmental Performance, and Environmental Disclosure of China" deals with the determinants of carbon disclosure information on the Chinese market. The paper could be of local interest to researchers on the Chinese market, however it has several flaws making it unsuitable from publication in this current form. I urge the authors to consider the following comments and improve the paper for potential future endeavours. Response to the reviewer The authors are thankful to the reviewer for devoting valuable time to provide us such useful insights. We believe that these suggestions were very useful in improving the quality of the manuscript.

Comment # 2 On page 2, lines 58-61 authors have provided a very confused discussion on some questions regarding venture capital or private equity etc. What do they mean exactly? Are these questions examined on this study? As I kept reading it seemed that author do not examine such issues mentioned on the introduction so why are they put forth at the first place? Response to the reviewer Thank you very much for your comments. These questions supported the above statement, such as the media’s impact on private equity or venture capital and social awareness implications through media reporting. However, to reduce the confusion, we have removed the question marks.

Comment # 3 There is not sufficient justification and theoretical background for the scope of the study. This fact makes the contribution of the study very limited to existing literature, even within chinese studies. Response to the reviewer Thank you very much for your valuable suggestion. We have improved the introduction part, including the theoretical background for the scope of the study.

Comment # 4 On page 3, lines 124-125, authors say that they focus on SOEs and non-SOEs as a potential determinant of carbon disclosures. Yet there is not explanation why authors expect such difference at the first place. There is no justification as to why state ownership or not affects CDI. Response to the reviewer Thank you very much for your comments. We have modified the statements as below “Still, a study is needed to understand CDI’s systematic influence on state-owned enterprises (SOE) and non-state-owned enterprises (NSOE), primarily because of the CDI quality effects on various stakeholders’ market competitiveness position.”

Comment # 5 All research hypotheses are not substantiated and even H2 and H3 are not derived from studies or theories related to the research question. Also as hypothesis are stated is very difficult to test and infer if they are valid or not. For example when authors say that CDI has a negative impacrt on enterprises, what "impact" they refer to? They need to be specific. Also the discussion leading to H1 is not supported by other studies. Simillarly, H4 and H5 refer to "benefitis" and "damage" to the corporation. What benefit and what damage? Overall the hypothesis development section needs a lot of work in order to be sound and substantiated. Response to the reviewer Thank you very much for your comments. The intended hypotheses are related to the gap in the literature. Second thing, we have explained hypotheses in a comprehensible format. Third, CDI has a negative impact based on the economic point of view. Fourth, according to your suggestion, we have restructured hypotheses with the explanation.

Comment # 6 On section 3.1 how positive reporting is separated from neutral? If a firm is following existing environmental rules and regulations isn't it positive? Why s deemed neutral? This distinction requires further consideration. Response to the reviewer Thank you very much for your comments. We have already mentioned that we have arranged data from different sources. The primary data source is related to CNRDs. Every year CNRD conducts a survey. They have created a separate question for the number of media reports as neutral, negative, and positive. We have already explained the impact of positive media reporting “The estimated results explain that media’s positive reporting offsets the additional benefits of stakeholders.”

Comment # 7 On page 6, on the paragraph of CID data collection, there are several metrics that are not presented on the first table as suggested. Also this part requires more justification from related literature. Response to the reviewer Thank you very much for your comments. We have tried to minimize the repetition, so we have explained the CID data collection paragraph metrics. Second, according to your suggestion, we have included justification from the literature to support the study’s scope.

Comment # 8 Since CDI is a dummy variable authors should have employed logistic regression analysis and not panel as performed. This is a serious issue that may affect results and conclusions. Response to the reviewer Thank you very much for your valuable suggestion. We have followed the previously published paper methodology “External Pressure, Corporate Governance, and Voluntary Carbon Disclosure: Evidence from China,” along with employing the Hausman specification test to detect the endogenous regressor in a regression model. See table 7.

Reviewer 2 Report

This is an interesting article. However, the authors need to clarify the contribution of the study.

The authors claim that their study explain that media’s positive reporting offsets the additional benefits of stakeholders. However, it may be subject to a specific country and media freedom depends significantly.

The English editing is needed too.

Author Response

Comment # 1 This is an interesting article. However, the authors need to clarify the contribution of the study. Response to the reviewer Thank you very much for your interest. We have clarified most of the parts where the clarification is needed

Comment # 2 The authors claim that their study explain that media’s positive reporting offsets the additional benefits of stakeholders. However, it may be subject to a specific country and media freedom depends significantly. Response to the reviewer Thank you very much for your suggestion. The current study highlights the impact of CDI influences either economic gain or loss of enterprises’ competitive position or financial value, or social awareness. Second, current study data availability and previously published relevant papers highlight media freedom significantly.

Comment # 3 The English editing is needed too. Response to the reviewer The current version of the manuscript has been reviewed by a native speaker with good knowledge of the subject.

Reviewer 3 Report

Although I have no problems with the paper being accepted as it is, below I will provide some comments that may help the authors improving further the paper. Please note that I don't feel qualified to judge about the English language and style, and this can be a issue to deal with as well.

The article is a fruitful contribution to the literature. Methodology is appropriate for the purposes of the study. The style in which the paper is written is certainly clear and concise. Argument is coherent.

In the section on relevant studies, the authors should provide a little more depth about the empirical literature then what is the case with the material they have in the literature review section. This is particularly the case regarding research on carbon disclosure in China. The authors could enrich their study by using the existing literature (for some recent studies on carbon disclosure in China, see references below). Although this would add some complexity to the paper, it will enhance the paper’s depth and provide additional data to refer to in terms of the analysis of the data and future research.

The authors use the literature without a real theoretical analysis. The paper would benefit from explicit use of a theoretical framework, such as a media-agenda setting theory and/or a legitimacy theory perspective (please see the references below). This would allow the authors to reflect more fully at the end of the article on the implications for theorizing about the topic, which is lacking.

The structure of the paper can be improved. One way of doing this is restructuring as the paper as follows: (i) Introduction; (ii) Theoretical literature review; (iv) Empirical literature review and hypotheses development; (v) Research design; (vi) Empirical results and discussion; and (vii) Summary and conclusion. Another way is: (i) Introduction; (ii) Relevant studies; (iii) Theory and hypotheses development; (v) Research design; (vi) Empirical results and discussion; and (vii) Summary and conclusion. Either way, the development of hypotheses should not be included in a literature review section.

I believe that the authors could gain some useful insights on how to start updating the references by using the following papers:

Brown, N., & Deegan, C. (1998). The public disclosure of environmental performance information—a dual test of media agenda setting theory and legitimacy theory. Accounting and business research, 29(1), 21-41.

Deephouse, D. L. (2000). Media reputation as a strategic resource: An integration of mass communication and resource-based theories. Journal of management, 26(6), 1091-1112.

Doan, M. H., & Sassen, R. (2020). The relationship between environmental performance and environmental disclosure: A meta‐analysis. Journal of Industrial Ecology, 24(5), 1140-1157.

Islam, M. A., & Deegan, C. (2010). Media pressures and corporate disclosure of social responsibility performance information: a study of two global clothing and sports retail companies. Accounting and business research, 40(2), 131-148.

Li, D., Huang, M., Ren, S., Chen, X., & Ning, L. (2018). Environmental legitimacy, green innovation, and corporate carbon disclosure: Evidence from CDP China 100. Journal of Business Ethics, 150(4), 1089-1104.

Li, L., Liu, Q., Tang, D., & Xiong, J. (2017). Media reporting, carbon information disclosure, and the cost of equity financing: evidence from China. Environmental Science and Pollution Research, 24(10), 9447-9459.

Patten, D. M. (2002). Media exposure, public policy pressure, and environmental disclosure: An examination of the impact of tri data availability. Accounting forum. 26(2), 152-171.

Pellegrino, C., & Lodhia, S. (2012). Climate change accounting and the Australian mining industry: exploring the links between corporate disclosure and the generation of legitimacy. Journal of Cleaner Production, 36, 68-82.

Pollach, I. (2014). Corporate environmental reporting and news coverage of environmental issues: An agenda‐setting perspective. Business Strategy and the Environment, 23(5), 349-360.

Watson, S. (2011). Conflict diamonds, legitimacy and media agenda: an examination of annual report disclosures. Meditari Accountancy Research, 19(1/2), 94.

Yu, H. C., Kuo, L., & Ma, B. (2020). The drivers of carbon disclosure: evidence from china’s sustainability plans. Carbon Management, 11(4), 399-414.

Author Response

Comment # 1

Although I have no problems with the paper being accepted as it is, below I will provide some comments that may help the authors improving further the paper. Please note that I don't feel qualified to judge about the English language and style, and this can be a issue to deal with as well. The article is a fruitful contribution to the literature. Methodology is appropriate for the purposes of the study. The style in which the paper is written is certainly clear and concise. Argument is coherent.

Response to the reviewer

Thank you very much for your positive response. We have tried our best to improve the manuscript’s English, where we did find grammatical mistakes.

Comment # 2

In the section on relevant studies, the authors should provide a little more depth about the empirical literature then what is the case with the material they have in the literature review section. This is particularly the case regarding research on carbon disclosure in China. The authors could enrich their study by using the existing literature (for some recent studies on carbon disclosure in China, see references below). Although this would add some complexity to the paper, it will enhance the paper’s depth and provide additional data to refer to in terms of the analysis of the data and future research.

Response to the reviewer

Thank you very much for your suggestion. We have added relevant material in the literature review part with updated published papers according to your suggestion. Second, we have followed your mentioned references to enhance the paper’s depth.

Comment # 3

The authors use the literature without a real theoretical analysis. The paper would benefit from explicit use of a theoretical framework, such as a media-agenda setting theory and/or a legitimacy theory perspective (please see the references below). This would allow the authors to reflect more fully at the end of the article on the implications for theorizing about the topic, which is lacking.

Response to the reviewer

Thank you very much. We have enhanced the theoretical framework with an explanation of media agenda setting theory (Brown, N., & Deegan, C. 1998) legitimacy theory perspective “(Pollach, I. 2014) reported that enterprises environmental agenda and media agenda are like a mirror of each other, while some impacts are different, but not vice versa”.

Comment # 4

The structure of the paper can be improved. One way of doing this is restructuring as the paper as follows: (i) Introduction; (ii) Theoretical literature review; (iv) Empirical literature review and hypotheses development; (v) Research design; (vi) Empirical results and discussion; and (vii) Summary and conclusion. Another way is: (i) Introduction; (ii) Relevant studies; (iii) Theory and hypotheses development; (v) Research design; (vi) Empirical results and discussion; and (vii) Summary and conclusion. Either way, the development of hypotheses should not be included in a literature review section.

Response to the reviewer

Thank you very much for your valuable suggestions. We have restructured papers, as suggested. (i) Introduction; (ii) Theoretical literature review; (iv) Empirical literature review and hypotheses development; (v) Research design; (vi) Empirical results and discussion; and (vii) Summary and conclusion.

Comment # 5

I believe that the authors could gain some useful insights on how to start updating the references by using the following papers:

Brown, N., & Deegan, C. (1998). The public disclosure of environmental performance information—a dual test of media agenda setting theory and legitimacy theory. Accounting and business research, 29(1), 21-41.

Deephouse, D. L. (2000). Media reputation as a strategic resource: An integration of mass communication and resource-based theories. Journal of management, 26(6), 1091-1112.

Doan, M. H., & Sassen, R. (2020). The relationship between environmental performance and environmental disclosure: A meta‐analysis. Journal of Industrial Ecology, 24(5), 1140-1157.

Islam, M. A., & Deegan, C. (2010). Media pressures and corporate disclosure of social responsibility performance information: a study of two global clothing and sports retail companies. Accounting and business research, 40(2), 131-148.

Li, D., Huang, M., Ren, S., Chen, X., & Ning, L. (2018). Environmental legitimacy, green innovation, and corporate carbon disclosure: Evidence from CDP China 100. Journal of Business Ethics, 150(4), 1089-1104.

Li, L., Liu, Q., Tang, D., & Xiong, J. (2017). Media reporting, carbon information disclosure, and the cost of equity financing: evidence from China. Environmental Science and Pollution Research, 24(10), 9447-9459.

Patten, D. M. (2002). Media exposure, public policy pressure, and environmental disclosure: An examination of the impact of tri data availability. Accounting forum. 26(2), 152-171.

Pellegrino, C., & Lodhia, S. (2012). Climate change accounting and the Australian mining industry: exploring the links between corporate disclosure and the generation of legitimacy. Journal of Cleaner Production, 36, 68-82.

Pollach, I. (2014). Corporate environmental reporting and news coverage of environmental issues: An agenda‐setting perspective. Business Strategy and the Environment, 23(5), 349-360.

Watson, S. (2011). Conflict diamonds, legitimacy and media agenda: an examination of annual report disclosures. Meditari Accountancy Research, 19(1/2), 94.

Yu, H. C., Kuo, L., & Ma, B. (2020). The drivers of carbon disclosure: evidence from china’s sustainability plans. Carbon Management, 11(4), 399-414.

Response to the reviewer

Thank you very much for your suggestion. We have updated our references accordingly. Precisely we have explained and cited more relevant references which you have suggested. We acknowledge the reviewer for these valuable references’ recommendation.

Reviewer 4 Report

Main Comments and Suggestions

The story is not clear; the abstract and introduction are not well written. Also in the introduction, you should clarify the contributions of the paper which are not elaborated well in the current paper. You can talk about the following contributions: What insights can you provide based on your finding? Do they push forward our understanding? What should we do with your research? Do you have any suggestions to improve the current regulation or practice? Adding the above discussion and extend your literature review may help you make more contributions and position your contributions better.

The paper seems to claim causality but does not discuss the potential endogeneity issue and its remedies sufficiently. See Li 2016, Endogeneity in CEO power: A survey and experiment, Investment Analysts Journal, 45 (3): 149-162 for a summary of methods to deal with the endogeneity problem. No need to use all these methods but at least discuss them in your scenario.

Additionally, you should refer to more recent development in this area. For instance, Li, F., Young, B., Morris, T. 2019. Corporate Visibility in Print Media and Corporate Social Responsibility. Sustainability forthcoming. Link your media reporting results to the literature more tightly.

Minor Comments and Suggestions

There are many typos, grammatical mistakes and awkward sentences throughout the paper, making it hard to read and understand. Try to avoid long sentences and vague words. Use short, precise, and concise sentences and be more straightforward. The last section of conclusion should summarize all your findings, their implications to researchers and practitioners, future direction for research, limitation of the current study, etc. You need to seriously proofread the paper and extend and update your references.

In conclusion, I would like to thank the authors for a very interesting, unique and potentially important paper. Hope these comments and suggestions can help further their study.

Author Response

Comment # 1

The story is not clear; the abstract and introduction are not well written. Also in the introduction, you should clarify the contributions of the paper which are not elaborated well in the current paper. You can talk about the following contributions: What insights can you provide based on your finding? Do they push forward our understanding? What should we do with your research? Do you have any suggestions to improve the current regulation or practice? Adding the above discussion and extend your literature review may help you make more contributions and position your contributions better.

Response to the reviewer

Thank you very much for your comments. We have modified the indicated part based upon the improvement of the manuscript. We believe that the research also provides insight based on the financial gains and losses of SOEs and NSOEs caused by CDI level reporting as media reports increase CDI’s influence on a budget of equity financing to promote or inhibit the value. For this reason, the study is needed to explain the relationship between CDI, Media reports, State or non-state pressure, and financial depression on shareholders to make policy recommendations for the welfare of environmental protection and sustainable development.

Comment # 2

The paper seems to claim causality but does not discuss the potential endogeneity issue and its remedies sufficiently. See Li 2016, Endogeneity in CEO power: A survey and experiment, Investment Analysts Journal, 45 (3): 149-162 for a summary of methods to deal with the endogeneity problem. No need to use all these methods but at least discuss them in your scenario.

Response to the reviewer

Thank you very much for your comments. We have explained the endogeneity problem in the manuscript with citations as follows.

“Arguably, the endogeneity problem can exist in the above described model setting [55]. Following the above model setting as a preliminary experiment, we did employ the Hausman specification test to detect the endogenous regressor in a regression model.”.

Comment # 3

Additionally, you should refer to more recent development in this area. For instance, Li, F., Young, B., Morris, T. 2019. Corporate Visibility in Print Media and Corporate Social Responsibility. Sustainability forthcoming. Link your media reporting results to the literature more tightly.

 Response to the reviewer

Thank you very much for your comments. We have explained more media role in manuscript with citation “Moreover, media coverage and visibility may increase the exhibition of enterprises’ supplementary claims [54].” in manuscript with citation. We think that reference is very supportive of our intended study.

Comment # 4

There are many typos, grammatical mistakes and awkward sentences throughout the paper, making it hard to read and understand. Try to avoid long sentences and vague words. Use short, precise, and concise sentences and be more straightforward. The last section of conclusion should summarize all your findings, their implications to researchers and practitioners, future direction for research, limitation of the current study, etc. You need to seriously proofread the paper and extend and update your references.

Response to the reviewer

Thank you very much for your comments. We have thoroughly revised our manuscript to shorten the sentences and reduce grammatical mistakes. Second, we have also revised the conclusion part carefully, including findings, future directions for research, and limitations so on. We believe that our manuscript version looks better than the previous version.

Comment # 5

In conclusion, I would like to thank the authors for a very interesting, unique and potentially important paper. Hope these comments and suggestions can help further their study.

Response to the reviewer

Thank you very much for your appreciation and interest. We have tried our best by acting upon your valuable suggestions.

Round 2

Reviewer 1 Report

I would like to thank the authors for their effort to address the majority of the comments raised. I believe the paper has been improved significantly relative to its initial form. A few editing and syntax mistakes are in some parts of the text, so authors need to re-read and improve the language accordingly.

Reviewer 4 Report

Well done. Big congrats!